# Novel Environmentally Friendly Covalent Organic Framework/Polylactic Acid Composite Material with High Chemical Stability for Sand-Control Material

**DOI:** 10.3390/polym15071659

**Published:** 2023-03-27

**Authors:** Wanjia Yang, Yongling Du, Benli Liu

**Affiliations:** 1Laboratory of Ecological Safety and Sustainable Development in Arid Lands, Northwest Institute of Eco-Environment and Resources, Chinese Academy of Sciences, Lanzhou 730000, China; yangwanjia@bjfu.edu.cn; 2University of Chinese Academy of Sciences, Beijing 100029, China; 3College of Chemistry and Chemical Engineering, Lanzhou University, Lanzhou 730000, China; duyl@lzu.edu.cn; 4Southern Marine Science and Engineering Guangdong Laboratory (Guangzhou), Guangzhou 511458, China

**Keywords:** covalent organic framework, polylactic acid, sand control materials, thermal stability, environmentally friendly

## Abstract

A new high-strength, thermally stable, and degradable covalent organic framework (COF) -modified polylactic acid fiber (PLA) material (COF-PLA) was constructed for reinforcing the PLA material, to be used to produce environmentally friendly sand barriers. The micrographs, structure, thermal stability, and photodegradation products of COF-PLA were investigated. The results indicated that the COF material was compatible with PLA, and that the COF-PLA material took on the merits of the COF, so that it had a more regular arrangement, smoother surface, and smaller size, and was more thermostable than PLA alone. The successful incorporation of the COF improved the thermal stability of PLA. The initial pyrolysis temperature of the COF-PLA material is 313.7 °C, higher than that of the PLA material at 297.5 °C. The photodegradation products of COF-PLA and PLA indicated that the COF and PLA materials were mixed in a complex manner. After photodegradation, the COF-PLA material can produce melamine molecules that can neutralize the lactic acid and CO_2_ produced by PLA, which can maintain the acid–base balance in sandy soil and is beneficial to plant growth. Therefore, COF-PLA degradation does not cause pollution, making it a promising sand-control material.

## 1. Introduction

Land desertification is an environmental and ecological issue, and one of the greatest threats to humanity in the 21st century [1]. Approximately 1/4 of the earth’s land, including hundreds of countries and regions, is currently affected by desertification. Desertification destroys natural environments and land productivity, causing great economic losses and threats to human survival and development [2]. China is among the countries seriously affected by land desertification [3]. Most land in China is affected by wind-erosion desertification, which accounts for 69.82% of the country’s desertified land area [2]. Therefore, it is important to implement appropriate measures to control desertification.

Various plant, mechanical, and chemical measures have been used to combat desertification. Mechanical sand-control measures, such as sand barriers, are usually used in harsh deserts where plants struggle to survive. These methods can stop the forward movement of sand dunes, stabilize the shifting sands, and create stable terrain for plants to grow [4]. Mechanical sand barriers are a common measure used worldwide. For example, straw-checker barriers are an effective and economical method and play an important role in controlling wind-blown sand in North China [5].

Many studies have been reported on mechanical sand barriers [6], including ways to set them up [7] and their protective benefits [8]. Polylactic acid (PLA) is a new biodegradable polymer made from starch-containing crops such as wheat, corn, sweet potatoes, potatoes, and sugar beets. PLA degrades into CO_2_ and H_2_O, and therefore does not cause secondary pollution to the environment [9,10]. Compared to traditional sand barriers, such as wheat straw, clay, and gravel, PLA sand barriers have many advantages, including being more environmentally friendly, lighter, easier to transport, more convenient to use, and less susceptible to weathering. For those reasons, they have great potential for extended applications [11].

However, the development and application of PLA are limited due to its high price, unstable mechanical properties, and low thermal stability, which are in urgent need of improvement [12,13]. Compounding other plant fibers with PLA can help overcome some of these limitations as plant fibers have strong mechanical properties which they can lend to the polymer [14,15]. These composites of PLA and plant fibers are also biodegradable, maintaining the PLA’s environmental-friendliness [16,17]. 

Therefore, the modification of PLA with plant fibers can not only improve the mechanical properties of PLA products, but also reduce their cost while maintaining their reported benefits. These composites can greatly increase the scope of PLA applications, and many studies have focused on preparing PLA-based composites using natural plant fibers as reinforcements. Ghazali et al. [18] prepared flax/PLA composites using a lamination method and investigated the effect of the fiber content on the tensile properties of the composites. It was found that the tensile strength and elasticity modulus first increased until fiber content reached 30 wt%, then decreased with further increasing fiber content. The maximum tensile strength was 101.6 MPa, 90% higher than that of PLA (53.9 MPa). Manshor [19] prepared durian peel fiber/PLA composites by injection molding and investigated the effects of the fiber content and alkaline treatment on the impact resistance and thermal properties of the composites. Their tests revealed that the impact resistance strength of the composites showed a decreasing–increasing–decreasing trend with an increase in fiber content. The best results were obtained when the fiber content reached 30 wt%. The impact resistance strengths of the composites exceeded those of PLA, and thermogravimetric analysis showed that the alkali pretreatment enhanced the thermal stability of the composites. Qian et al. [20] prepared cotton fiber/PLA composites using melt-blending and injection-molding methods and investigated the effects of different fiber content and lengths. The results indicated that the tensile strength and elastic moduli of the composite were maximized at a fiber content of 20 wt%, and they increased by 35.7% and 44.9%, respectively, compared with those of pure PLA. Sun et al. [21] prepared loofah fiber bundle/PLA composites by a hot-pressing method using a loofah fiber bundle as reinforcement and analyzed the effect of fiber content and length on the tensile properties of the composites. The results showed that a fiber content of 5 wt% and length of 40 mm had the best reinforcement effect.

However, plant fiber, or cellulose, is usually produced by chemical extractions, such as the alkali cooking method with the chemical extraction process, which has many extraction steps and may cause chemical contamination during production [22,23]. Covalent organic frameworks (COFs) are a new category of low-cost, structurally stable porous crystalline polymers which can be prepared more simply [24].

Bhadra et al. [25] found that novel, cost-effective, and moderately crystalline COF materials can be prepared by a simple one-step method using a dimethylacetamide(DMAc): dimethyl sulfoxide(DMSO) (2:1) solvent combination. Moreover, due to the irreversible nature of the β-ketoenamine structure formation in the reaction, the reaction process is free of derivatives and environmentally friendly. COFs are also durable and exhibit highly ordered, pre-engineered, and functionalized porous structures with high chemical stability. Due to their fully organic nature, they are expected to be compatible with organic polymeric substrates [26].

In recent years, COFs have been widely used in storage adsorption [27,28], hydrogen production [29,30], separation [31], catalysis [25,31], optoelectronics [32], and hybrid membranes [26]. COF (TpTt), which has triazine and a β-ketoenamine structure, is produced by reacting melamine (Tt) with 2,4,6-tricarboxyresorcinol (Tp) aldehyde. The triazine core promotes strong π–π covalent bonds, E-olefins or carbonyl C=O interaction with unsaturated bonds, and the reaction between the amino group of melamine and the aldehyde group of 2,4,6-tricarboxyresorcinol that produces the β-ketoenamine structure [25]. Due to the irreversible nature of the formation of the β-ketoenamine structure, the new triazine-functionalized hybrid COF shows high chemical stability even under light irradiation. COF (TpTt) has triazine and a β-ketoenamine structure, produced by reacting melamine (Tt) with 2,4,6-tricarboxyresorcinol (Tp) aldehyde. The triazine core can promote strong π–π covalent bonds interacting with unsaturated bonds, E-olefins, or carbonyl C=O, and the reaction between the amino group of melamine and the aldehyde group of 2,4,6-tricarboxyresorcinol can produce the β-ketoenamine structure [25]. Due to the irreversible property of the formation process of the β-ketoenamine structure, the new triazine-functionalized hybrid COF shows high chemical stability even under light irradiation.

In this study, chemically stable and cost-effective COF materials were prepared using a one-step method, and the micrographs, structure, thermal stability, and photodegradation of PLA modified by a highly structured COF material were studied.

## 2. Materials and Methods

### 2.1. Materials

Lactic acid (AR, Jiangsu Sanmu Group Chemical Plant, Jiang Su, China). Zinc oxide (AR, Chengdu Colon Chemicals Co., Ltd., Sichuan, China). Ethyl acetate (AR, Keanlong Bohua Pharmaceutical Chemical Co., Ltd., Tianjin, China). Stannous octoate (Shanghai Maclean Biochemical Technology Shares Co., Ltd., Shanghai, China). Trichloromethane (AR, Lianlong Bohua Pharmaceutical Chemical Co., Ltd., Tianjin, China). Methyl alcohol (AR, Lianlong Bohua Pharmaceutical Chemical Co., Ltd., Tianjin, China). Melamine/1,3,5-Triazine-2,4,6-triamine (Tt, Chengdu Colon Chemicals Co., Ltd., Chengdu, China). 2,4,6-Triformylphloroglucinol aldehyde (Tp, Teng Qian Biotechnology Co., Ltd., Shanghai, China). Acetic acid (Hac, AR, Jiangsu Sanmu Group Chemical Plant, Jiang Su, China). α-cellulose (25μm, Shanghai Maclean Biochemical Technology shares Co., Ltd., Shanghai, China). Dimethylacetamide (DMAc, Shanghai Maclean Biochemical Technology Shares Co., Ltd., Shanghai, China). Dimethylsulfoxide (DMSO, Shanghai Maclean Biochemical Technology Shares Co., Ltd., Shanghai, China).

### 2.2. Instruments and Equipment

Fourier transform infrared spectrometer (NEXUS 670, Nicolet Instruments Inc., Missouri, TX, USA). Thermal gravimetric analyzer (TGA 5500, TA Instruments Inc., New Castle, DE, USA). Circulating water multi-purpose vacuum pump (SHB-III, Zhengzhou Great Wall Technology and Trade Co., Ltd., Henan, China). Ampere bottle (Synthware Glass Instrument Co., Ltd., Beijing, China). Vacuum drying oven (DZF-6020, Shanghai Hongdu Electronic Technology Co., Ltd., Shanghai, China). Thermo Exactive-GC mass spectrometer (Beijing Jingkerida Technology Co., Ltd., Beijing, China). Brunauer–Emmett–Teller automatic surface and porosity analyzer (BET-ASAP2020-HD4.00, Micromeritics Instrument Corp, Norcross, GA, USA). X-Ray Diffraction, XRD (Rigaku D/max-2400, Beijing Brock Technology Co., Ltd., Beijing, China).

### 2.3. Samples Preparation

#### 2.3.1. Synthesis of Polylactic Acid (PLA)

First, 200 mL of lactic acid was distilled to dehydration in a vacuum pump under reduced pressure at 100 °C. Zinc oxide was dried at 500 °C and then added to produce a 1% by volume solution in the lactic acid. The temperature was then adjusted to 150 °C and the mixture was distilled for 3 h to yield lactide. Next, 5 g lactide was dissolved in 10 mL ethyl acetate at 45 °C, then cooled at room temperature to obtain pure lactide with a yield of approximately 75%.

Next, pure lactide was vacuumed and frozen in an amperometric flask. Stannous octanoate with a mass fraction of approximately 0.1% lactide was added. The mixture was thawed slowly and then heated for thorough mixing. The mixture was taken through three cycles of thawing, vacuum, and a high-purity nitrogen gas exchange operation before the amperometric flask was sealed under vacuum conditions and placed in an oven at 220 °C for 48 h to produce the PLA. The resulting PLA is shown in Figure 1a.

#### 2.3.2. One-Step Preparation of Covalent Organic Frameworks (COFs)

At 150 °C, 38 mg of melamine (Tt), 63 mg of 2,4,6-triformylphloroglucinol aldehyde (Tp), 2 mL DAMc, 1 mL DMSO, and 0.3 mg of Hac (acetic acid) catalyst were placed in a test tube to react for 48 h. During this period, the aldehyde groups in phloroglucinol and trimylamine reacted to form a structurally stable reticular COF structure with multiple aldehyde groups. A schematic representation of the COF synthesis is shown in Figure 2.

#### 2.3.3. Synthesis of Covalent Organic Frameworks (COFs)-Modified PLA (COF-PLA)

One gram of PLA was dissolved in 50 mL of chloroform and refluxed to thoroughly dissolve the solid. Next, 2 mg of COF nanomaterial was added to the refluxing mixture. After the complete dissolution of the solid, the mixture was allowed to cool. The PLA-COF solution was added dropwise while stirring into 150 mL of methanol. After the separation of the solid product from the solution was observed, the flocculent product was washed with methanol and the resulting COF-PLA was obtained, as shown in Figure 1b. 

According to the solubility parameter close principle, the COF can be dissolved in chloroform since both COF and chloroform are polar. Additionally, the dissolution did not change the COF structure. For example, the structure of a polymer dissolved in toluene remains unchanged. In addition, COF affords ultrahigh chemostability in strong acid, alkali, and boiling water, as well as γ radiation [25]. Therefore, the COF can maintain its structural integrity during dissolution.

However, as COFs have a stable structure and large molecular weight, their solubility in chloroform is poor. Therefore, we prepared different COF-PLA materials with weight ratios (COF/PLA) of 1:300, 1:500, and 1:1000.

### 2.4. Test Methods

#### 2.4.1. Scanning Electron Microscopy (SEM) of the Synthetic Materials

Micrographs of the samples were obtained using scanning electron microscopy. The samples were sprayed with gold before imaging, and an acceleration voltage of 10 kV was applied.

#### 2.4.2. Brunauer–Emmett–Teller (BET) of COF 

Brunauer–Emmett–Teller (BET) analysis was used to determine the surface area (SBET) of the COF.

#### 2.4.3. Powder X-ray Diffraction (PXRD) of COF

The crystalline structures of the synthesized nanomaterials were identified using powder X-ray diffraction (PXRD).

#### 2.4.4. Fourier-Transform Infrared (FT-IR) Spectroscopic Evaluations of the Synthetic Materials

Fourier-transform infrared spectroscopy (FT-IR) was used to study the molecular structure and chemical composition of the samples in the range of 500–4500 cm^−1^.

#### 2.4.5. Thermo-Gravimetric Analysis (TGA) of the Synthetic Materials

The thermal decomposition process, thermal stability, and compositional changes in the samples were analyzed by TGA using a thermal analyzer. The temperature ranged from room temperature to 500 °C with a heating rate of 10 °C/min, a nitrogen flow rate of 30 mL/min, and a sample volume of approximately 5 mg.

#### 2.4.6. Photodegradation Experiments

First, 50 g of sand was washed, dried, and placed in a flat quartz boat. Second, 0.1 g PLA or 0.1 g COF-PLA composite material was dissolved in chloroform and slowly dripped onto the sand in the quartz boat. Subsequently, a stable layer of the PLA or COF-PLA was formed on the sand surface after the solvent evaporated. The PLA membrane was colorless and the COF-PLA membrane was light yellow. 

Next, the PLA and PLA-COF quartz boats were doused with a small amount of water and placed under a wired light source with a current of 10 A. As irradiation proceeded, the color of the PLA quartz boat became more yellow, and the COF-PLA quartz boat became lighter, indicating that both PLA and COF-PLA were degraded. Finally, the degradation products of the PLA and COF-PLA materials were analyzed using liquid mass spectrometry.

## 3. Results and Discussion

### 3.1. Microstructure

Micrographs of the synthesized materials are shown in Figure 3. Figure 3a shows that the PLA material has large, thick folds with a scattered arrangement and a clear banded distribution. Figure 3b shows that the COF material has a stable structure, uniform and orderly size distribution, and a highly ordered arrangement. Figure 3c shows that the COF-PLA material has banded morphology and a smooth surface, which indicates that the COF nanomaterials were successfully embedded in and are well compatible with the PLA material. While the COF-PLA material integrates the advantages of the stable structure of the COF material, it exhibits the same banded morphology as PLA, but with smaller, more orderly arranged bands, and a smoother surface.

### 3.2. Brunauer–Emmett–Teller (BET) Analysis of COF

The surface area (SBET) of the COF was determined by the Brunauer–Emmett–Teller automatic surface and porosity analyzer. The Brunauer–Emmett–Teller (BET) analysis of the COF revealed that it had a surface area (SBET) of 169.20 m^2^/g. 

### 3.3. Powder X-ray Diffraction (PXRD) of the Synthetic Materials

Figure 4 shows the powder X-ray diffraction patterns of the synthesized materials. The PXRD pattern of PLA reveals two main characteristic peaks at 2θ = 17.1° and 19.7°. Figure 4b shows that the 002 crystal peak of COF is at 2θ = 27.6. The characteristic peak of 2θ = 9.6° is attributed to the reflections from the 100 planes in COF. The reported PXRD data show similar results [25]. There are some small differences between the peaks in this paper and those previously reported; this is because small differences in reaction conditions, such as reaction temperature and reaction time, lead to small differences in the crystal lattice structure of the COF. In addition, small differences in starting materials, such as dosage and ratio of starting materials, also cause small changes in the crystal lattice structure of the COF. In conclusion, there are many factors that can affect the lattice structure. The PXRD pattern of COF-PLA displays three characteristic peaks at 2θ = 17.1°, 19.7°, and 27.6. This illustrates that the COF material was successfully mixed with the PLA. However, the characteristic peak at 2θ = 27.6° was not observed because the amount of COF used to modify the PLA material was small.

### 3.4. FT-IR Analysis of the Synthetic Materials

Figure 5 shows the FT-IR spectra of the synthesized materials. The FT-IR spectrum of PLA (Figure 5a) displays two main characteristic peaks at 3505 and 1044.4 cm^−1^, which correspond to -OH stretching and bending vibration peaks, respectively. The troughs at 2947 and 2998 cm^−1^ correspond to the -CH symmetric and asymmetric stretching vibration peaks, respectively. The peak at 1213 cm^−1^ was caused by the C-C stretching vibration. A C-O stretching vibration peak was observed at 1127 cm^−1^. The characteristic peak of the amorphous phase for crystalline is shown at 871.2 cm^−1^, and the peak at 756 cm^−1^ is caused by the crystalline phase. This spectrum is in agreement with the spectroscopic results of a previous study [25], confirming the successful preparation of PLA.

Figure 5b shows that the characteristic peak of the amino group on the ring-mounted structure of the COF material was at 3396 cm^−1^. The peak at 3210 cm^−1^ is characteristic of the hydroxyl group on the benzene ring. The peaks at 1615 and 1651 cm^−1^ were attributed to the C=O double bonds in the COF material. The vibration peak of the trimylamine ring group was observed at 832 cm^−1^, and the vibration peak at 808 cm^−1^ was attributed to the benzene ring. These results confirmed the successful synthesis of new COF materials containing both melamine and benzene ring structures.

As shown in Figure 5c, the FT-IR spectra of the newly produced COF-PLA material contain not only the characteristic vibration peaks of PLA, such as the OH stretching vibration peak at 3498 cm^−1^, amorphous phase correlation peak at 871 cm^−1^, and crystalline phase correlation peak at 756 cm^−1^, but also the vibration peaks of the new COF material, including the C=O double bond vibration peak at 1619 cm^−1^ and the benzene ring vibration peak at 810 cm^−1^. Hence, Figure 5c confirms the successful doping of PLA with the COF material.

### 3.5. Liquid Chromatography-Mass Analysis of the Synthetic Materials

Figure 6a,b show the total liquid mass spectra of PLA and COF-PLA, respectively, and many of the peaks in these spectra are randomly produced and therefore do not have any specific meaning. In addition to these spectra, the liquid mass spectrometer also provided a large number of high-resolution spectra. These high-resolution spectra contain truly meaningful peaks, but they are not shown here as they were large in numbers. We have only shown the randomly produced peaks (Figure 6a,b) in total liquid mass spectra while explaining all the highest peaks obtained in the high-resolution spectral spectrums. 

As shown in Figure 6a, after PLA was degraded, a fragment with a mass of 116 was formed by removing a stable C=O molecule from the two PLA structural units. The fragment with a mass of 160 was formed by the removal of a CH_3_-CH-C=O group from the PLA structural units and belongs to a metastable structure. The fragment with a mass of 182 was formed by removing a hydrogen proton and a sodium ion from the metastable structure mentioned above. The fragment with a mass of 455 was formed by removing a hydrogen proton from six PLA structural units and one sodium ion. A fragment with a mass of 504 was formed from seven PLA structural units. The fragment with a mass of 527 was composed of seven PLA monomers and one sodium ion, which belongs to a metastable structure. A fragment with a mass of 670 was formed from nine PLA structural units and one sodium ion, with a hydrogen proton removed. A fragment with a mass of 631 was formed by removing one oxygen atom from the nine PLA structural units belonging to the steady-state structure.

As shown in Figure 6b, the fragment with a mass of 126 corresponds to melamine in the COF material. The fragment with a mass of 160 was formed by the removal of one oxygen atom from two PLA monomers, which belong to a metastable structure. The fragment with a mass of 182 was formed by a melamine monomer and a PLA monomer with one oxygen atom removed, belonging to a metastable structure. The fragment with a mass of 209 was a stable structure formed by the removal of one hydrogen proton from a triformylphloroglucinol aldehyde in the COF. The fragment with a mass of 321 was a stable structure consisting of melamine and triformylphloroglucinol aldehyde, with one oxygen atom removed. The fragments with masses of 116, 160, 455, and 527 all correspond to the photodegradation products of PLA.

The photodegradation products of COF-PLA and PLA in the spectra confirmed that the COF and PLA materials were mixed in a complex manner. The degradation of COF-PLA resulted in the formation of melamine monomers that neutralized the lactic acid and carbon dioxide degraded from PLA. This is a useful balancing process because all stable melamines can be neutralized with PLA and CO_2_. The remaining PLA can be biodegraded and the remaining unstable CO_2_ can be absorbed by the atmosphere or plants. This process allows the acid–base balance of sandy soils to be maintained even as the material degrades, thus supporting plant growth. 

### 3.6. Thermal Stability Analysis of the Synthetic Materials

Figure 7a signifies the thermal analysis diagram of PLA. The initial pyrolysis temperature of PLA is 297.5 °C and the final complete elimination temperature of PLA is 801.5 °C.

As can be seen from Figure 7b, the new COFs slowly began to lose weight from zero temperature and lost about 30% of their components at 359 °C, which may indicate the process of intermolecular recombination and solidification of COF. During this process, the COF lost water and other micromolecules. This is in good agreement with previous reports which suggest the successful synthesis of a novel COF material with polyamino groups [25]. 

Figure 7c–e demonstrate that the thermal stability of COF-PLA with different weight ratios was better than that of pure PLA. The initial pyrolysis temperature of COF-PLA with different weight ratios was higher than that of pure PLA at 297.5 °C. This improved thermal stability was due to the interaction of the triazine core with the strong π–π covalent bond of carbonyl group C=O and the formation of β-ketoenamine links by the amino groups of melamine and the aldehyde groups of 2,4,6-triformylphloroglucinol aldehyde [25]. Therefore, the thermal stability of the PLA was improved by the successful incorporation of a structurally stable COF.

In addition, due to the low solubility of COF in chloroform, lower weight ratios between COF and PLA result in a better doping effect and, therefore, better thermal stability of COF-PLA. 

Furthermore, the presence of water vapor may lead to the production of nitrogen in the COF pyrolysis process. Nitrogen is an essential nutrient for plant growth, and this process may allow the COF-PLA to act as a fertilizer for sand-fixing plants while also contributing to sand fixation.

## 4. Conclusions

In this study, we prepared a highly chemically stable and cost-effective COF material through a one-step method, and modified PLA with this COF to prepare a new degradable COF-PLA material with good thermal stability for use in high-strength sand barriers. Compared to other materials used as sand barriers, COF-PLA materials are simple to prepare, pollution-free, and economical. Furthermore, the experimental results illustrated that the new COF-PLA material had a stable structure, regular arrangement, smooth surface, and strong heat resistance. During the degradation process, the COF-PLA produces a melamine monomer which can neutralize the lactic acid and carbon dioxide produced during the degradation of PLA. This may help maintain the acid–base balance between sandy soil and plant growth. This new COF-PLA material has great potential applications for sand fixation in desert regions.

## Figures and Tables

**Figure 1 polymers-15-01659-f001:**
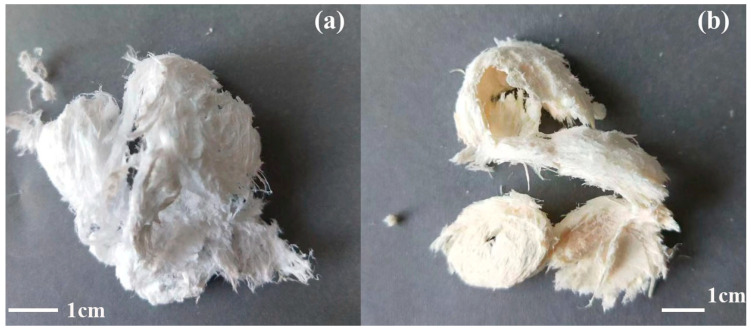
(**a**) Polylactic acid (PLA) and (**b**) covalent organic frameworks (COFs)-modified PLA (COF-PLA).

**Figure 2 polymers-15-01659-f002:**
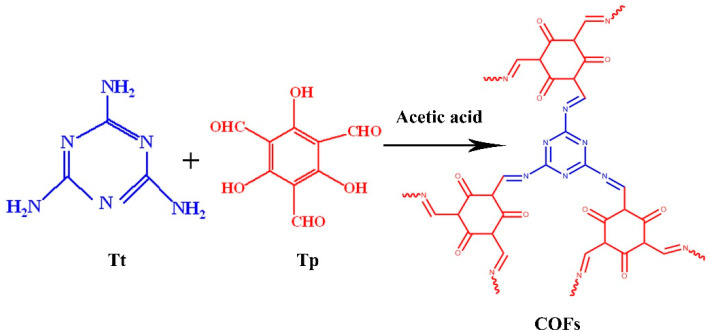
Schematic representation of the COF synthesis.

**Figure 3 polymers-15-01659-f003:**
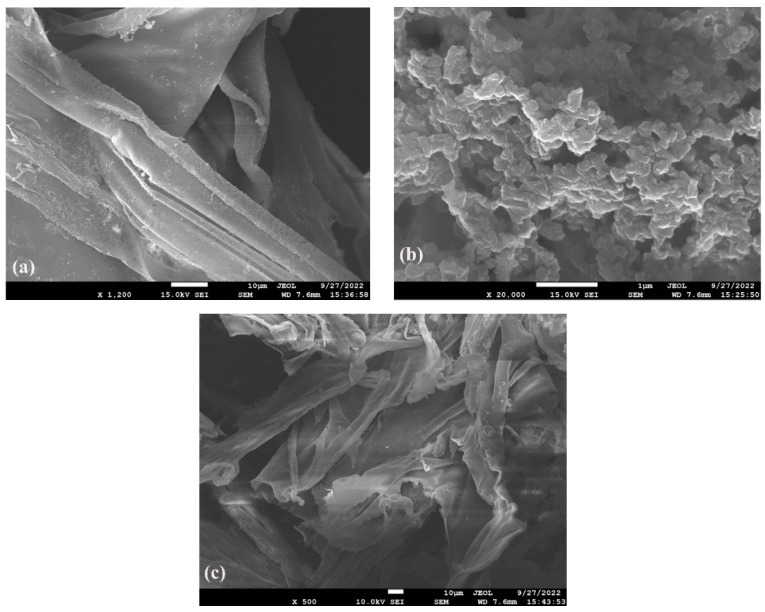
SEM images of (**a**) PLA, (**b**) COF, and (**c**) COF-PLA.

**Figure 4 polymers-15-01659-f004:**
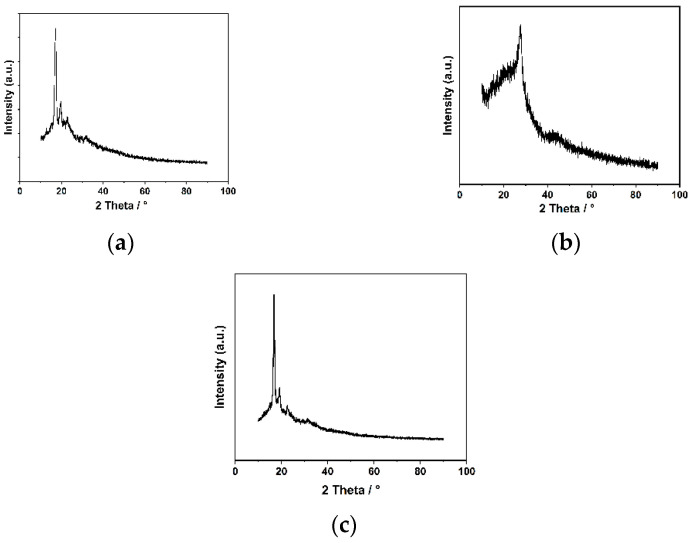
PXRD patterns of (**a**) PLA, (**b**) COF, and (**c**) COF-PLA.

**Figure 5 polymers-15-01659-f005:**
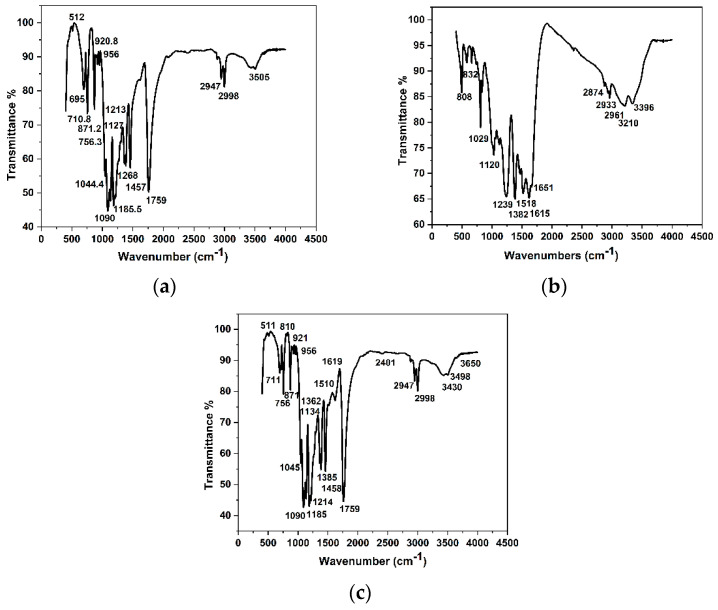
FT-IR of (**a**) PLA, (**b**) COFs, and (**c**) COFs-PLA.

**Figure 6 polymers-15-01659-f006:**
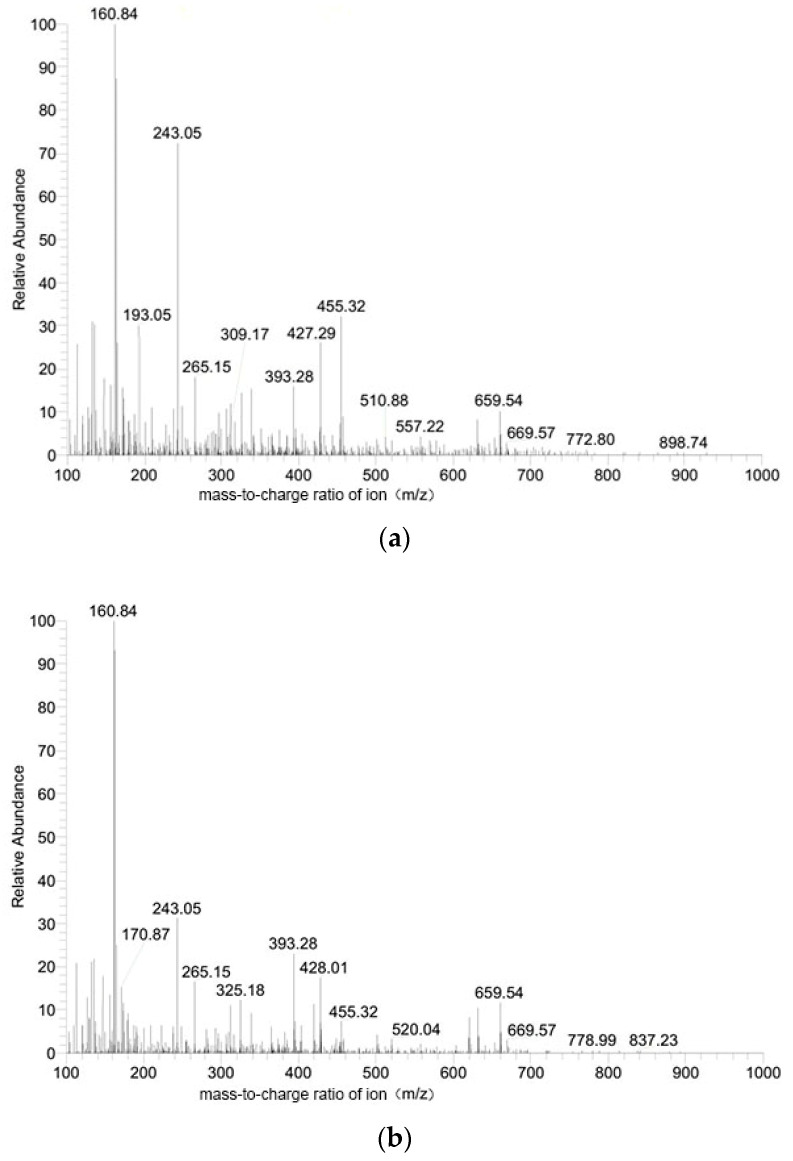
Liquid chromatography-mass spectrometry of the photodegradation productions of (**a**) PLA, (**b**) COF-PLA.

**Figure 7 polymers-15-01659-f007:**
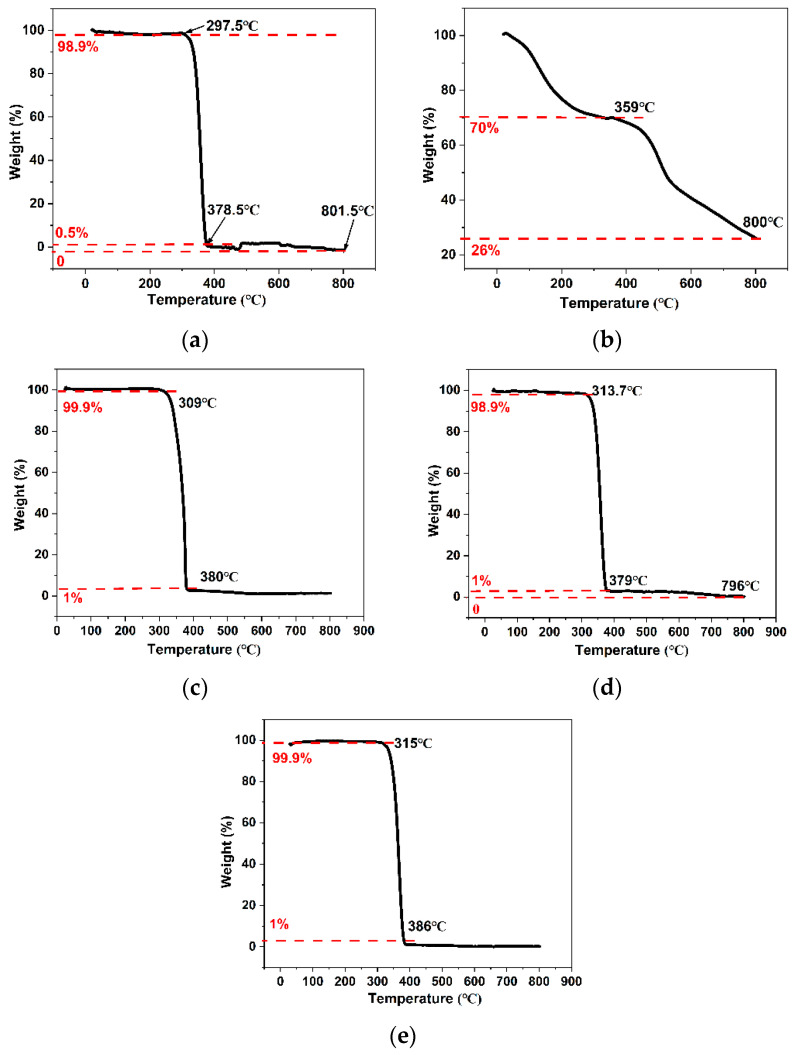
Thermogravimetric curves of (**a**) PLA, (**b**) COFs, (**c**) COF-PLA (COF/PLA: 1/300), (**d**) COF-PLA (COF/PLA: 1/500), and (**e**) COF-PLA (COF/PLA: 1/1000).

## Data Availability

The data presented in this study are available on request from the corresponding author.

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
