# Peer review of "Novel Environmentally Friendly Covalent Organic Framework/Polylactic Acid Composite Material with High Chemical Stability for Sand-Control Material"

_polymers, 2023, doi:10.3390/polym15071659_

Round 1

Reviewer 1 Report

The manuscript is well written and is of good scientific interest. However, it still needs significant editing to improve the paper's scientific or general quality.  Author is asked to respond to the following criticisms and incorporate necessary corrections in main text file.

1.    Please give the full form of DMAc: DMSO in line 90.

2.    Please cite some latest references in line 57-59, such as

https://doi.org/10.1016/j.biortech.2022.128255 (Used pine needles as reinforcement)

DOI: 10.1039/D1MA00429H (used hibiscus sabdariffa as reinforcing agent)

3.    In line 208, the author has mentioned “The troughs at 2998 cm-1 and 1268 cm-1 belong to -CH symmetric  and asymmetric stretching vibration peaks, respectively.” Please check the literature and FTIR data. Such a huge difference in C-H symmetric and asymmetric stretching values is not possible.

4.    Again, the peak at 1213 cm-1 cannot be assigned because of  -CH3 stretching. Please check the data and rewrite the section.

5.    Liquid chromatography-mass analysis of samples is not properly discussed. I think author should discuss the significance of all highly intense peaks in a better way. Here, author has just chosen the peaks randomly from the spectrum and discussed them, even though some of them are not clearly visible. Please comment.   

Reviewer 2 Report

The manuscript by Liu and co-worker entitled as “A novel environmentally friendly Covalent Organic Framework/Polylactic Acid composite material with high chemical stability for Sand control material” reports about the fabrication of a composite materials ((COF-PLA)) made of Covalent Organic Framework (COF) and polylactic acid fiber (PLA) material. The purpose of the modification of PLA with COF was to increase the thermal stability. Moreover the melamine molecules resulted from the photodegradation can also neutralize the lactic acid and CO2 produced by PLA, which can maintain the acid-base balance in sandy soil and is beneficial to plant growth. The concept of the present work seems to be interesting. However I have certain queries on structural characterization of pristine COFs and also the COF-PLA materials. The purpose of the COF incorporation with PLA is not considerable by the resulted experimental data. The manuscript should be modified thoroughly before its publication. Here are some points to be considered.

1.     The authors did not produce any PXRD of the mentioned COF. In order to claim it as COF, the author should examine the PXRD pattern of the materials. The PXRD of the COF should be also compared with the reported PXRD data.

2.     It is also suggested to measure the BET surface area of the COF.

3.     As described, the synthesis of COF-modified PLA (COF-PLA) involved the complete dissolution of pristine COF. How was the polymeric COFs dissolved in chloroform? How was the structural integrity of the COF maintained during the dissolution?

4.     The weight ratio between COF and PLA is 1:500 (from the synthesis procedure: 1 g of PLA and 2 of mg COF). Please explain how this low amount of COF will reinforce the PLA material.

5.     The authors also claim that the photodegradation of COF-PLA will produce melamine molecules which can neutralize the lactic acid and CO2 produced by PLA. Are there sufficient melamine molecules for the neutralization? (as the weight ratio of COF and PLA was very low)

6.     The PXRD of the COF-PLA material should be compared with the pristine COF in order to check the structural integrity of the pristine COF in COF-PLA material.

7.     The authors showed that the successful incorporation of COF improved the thermal stability of PLA (PLA material at 297.5℃ → COF-PLA material at 313.7℃). The increase of the thermal stability is not much considerable. Is it possible to increase the thermal stability to higher extent by increasing the amount of COF?

Round 2

Reviewer 1 Report

Author has incorporated necessary corrections in revised version and thus may be accepted in its current form.

Author Response

Thank you for the positive comments to this manuscript. We have checked the languate again.

Reviewer 2 Report

Although the authors have revised the manuscript by considering the mentioned points, however the structural characterization of the COF is still missing. The PXRD data has been included in the revised manuscript but the PXRD data has not been compared together with the reported data for the same COF. The peak seems to be at higher 2θ, which is different compared to the reported COF, the peak may arise from either starting materials or any other byproduct. The comparison of PXRD should be included in the revised manuscript.
